# Assessment of Acute Lesions of the Biceps Pulley in Patients with Traumatic Shoulder Dislocation Using MR Imaging

**DOI:** 10.3390/diagnostics12102345

**Published:** 2022-09-28

**Authors:** Georg C. Feuerriegel, Nicolas S. Lenhart, Yannik Leonhardt, Florian T. Gassert, Peter Biberthaler, Sebastian Siebenlist, Chlodwig Kirchhoff, Marcus R. Makowski, Klaus Woertler, Alexandra S. Gersing, Jan Neumann, Markus Wurm

**Affiliations:** 1Department of Radiology, Klinikum Rechts der Isar, School of Medicine, Technical University of Munich, 81675 Munich, Germany; 2Department of Trauma Surgery, Klinikum Rechts der Isar, School of Medicine, Technical University of Munich, 81675 Munich, Germany; 3Department of Orthopedic Sports Medicine, Klinikum Rechts der Isar, School of Medicine, Technical University of Munich, 81675 Munich, Germany; 4Musculoskeletal Radiology Section, Klinikum Rechts der Isar, School of Medicine, Technical University of Munich, 81675 Munich, Germany; 5Department of Neuroradiology, University Hospital of Munich, LMU Munich, 80333 Munich, Germany

**Keywords:** biceps pulley, shoulder dislocation, shoulder instability, rotator interval, pulley lesion, LHBT

## Abstract

Background: Shoulder dislocations represent common injuries and are often combined with rotator cuff tears and potentially damage to the biceps pulley. Purpose: To assess the occurrence and type of biceps pulley lesions in patients after traumatic anterior shoulder dislocation using 3T MRI. Methods: Thirty-three consecutive patients were enrolled between June 2021 and March 2022 (14 women, mean age 48.0 ± 19 years). All patients underwent MR imaging at 3 T within one week. Images were analyzed for the presence and type of pulley tears, subluxation/dislocation of the LHBT, rotator cuff lesions, joint effusion, labral lesions, and osseous defects. Results: Seventeen patients (52%) with traumatic anterior shoulder dislocation demonstrated biceps pulley lesions. Of those, eleven patients (33%) showed a combined tear of the sGHL and CHL. All seventeen patients with lesions of the biceps pulley showed associated partial tearing of the rotator cuff, whereas three patients showed an additional subluxation of the LHBT. Patients with pulley lesions after dislocations were significantly older than those without (mean age 52 ± 12 years vs. 44 ± 14 years, *p =* 0.023). Conclusion: Our results suggest an increased awareness for lesions of the biceps pulley in acute traumatic shoulder dislocation, particularly in patients over 45 years.

## 1. Introduction

Lesions of the biceps pulley are a common cause for anterior shoulder pain [1,2]. The biceps pulley is a capsuloligamentous complex which stabilizes the horizontal portion of the long head of the biceps tendon (LHBT) at the anterior aspect of the humerus and is mainly formed by the coracohumeral ligament (CHL), the superior glenohumeral ligament (sGHL) as well as fibers of the supraspinatus (SSP) und subscapularis (SSC) tendons [3,4,5,6,7]. Pulley lesions can be caused by degenerative changes, acute trauma, repetitive micro trauma or in association with rotator cuff tears [8,9]. Different biomechanical patterns have been described to cause a lesion or tear of the biceps pulley including a forcefully stopped overhead throwing motion, repetitive forceful internal rotation above the horizontal plane as well as a fall on the outstretched arm with full external or internal rotation of the humeral head [8,9,10]. Lesions of the biceps pulley can be associated with internal anterosuperior impingement as well as rotator cuff tears (in particular SSP and SSC lesions), lesions of the superior labrum anterior-posterior (SLAP), LHBT instability, and tears [1,2,4,8,11,12,13]. The subluxation or subluxation of the LHBT which is caused by an insufficient or ruptured biceps pulley is particularly associated with a higher prevalence of SSC and SSP lesions [13].

Anterior traumatic shoulder dislocations are most often caused by a forceful external rotation of the abducted arm which levers the humeral head out of its socket [14]. The extent of damage to the soft tissue is variable and may involve bony, cartilaginous, ligamentous as well as tendinous or muscular structures [15]. Most often, damage to the anteroinferior labro-ligamentous complex (Bankart lesion) is seen with or without osseous avulsion, damage to the capsule as well as a Hill-Sachs lesion [15,16]. So far, the role of the rotator interval and the biceps pulley in anterior shoulder instability remains unclear [17,18,19]. Thiesen et al. found that patients above 40 years showed a greater height and base of the rotator interval after anterior dislocation compared to a control group [17]. It was also shown that elderly patients in particular suffer from injuries to the rotator interval after shoulder dislocation [20,21,22]. To the best of our knowledge, direct damage to the biceps pulley in acute anterior shoulder dislocation has not been assessed yet. Therefore, the aim of our study was to assess the prevalence and degree of pulley lesions as well as lesions of the LHBT and rotator cuff using magnetic resonance imaging in patients after acute traumatic shoulder dislocations.

## 2. Material and Methods

### 2.1. Study Population and Ethics Statement

Patients presenting with acute traumatic shoulder dislocations (N = 33, 14 women, mean age 48.0 ± 19 years) admitted to the orthopedic and trauma surgery departments between June 2021 and March 2022 were enrolled in the study. Patients with dislocations due to chronic shoulder instability or complex fractures were excluded. All procedures were conducted according to the principles expressed in the Declaration of Helsinki. Written informed consent was obtained from all study participants prior to inclusion. The prospective analysis was approved by our institutional review board (Ethics Commission of the Medical Faculty, Technical University of Munich, Germany; Ethics proposal number 42/21S).

### 2.2. MR Imaging

After clinical examination and thorough anamnesis, each patient underwent a 3T MR imaging examination (Ingenia Elition; Philips Healthcare, Best, The Netherlands) of the shoulder using a dedicated 16-channel shoulder coil. A clinical routine imaging protocol was used including triplanar intermediate weighted (IM) turbo-spin echo (TSE) sequences with spectral fat saturation, a sagittal T2-weighted TSE sequence and a coronal T1 weighted TSE sequence. Detailed scan parameters are displayed in (Table 1). In cases where arthroscopy was performed, the intraoperative results were assessed and compared to the findings of MR imaging.

### 2.3. Treatment

Young and active patients suffering from first-time anterior shoulder dislocation routinely undergo arthroscopic refixation of the antero-inferior labrum at our hospital [23,24]. Concomitant pathologies can also be addressed during this operative intervention. Patients over 40 years of age with or without sportive demand are counseled either to opt for operative or conservative treatment depending on co-morbidities and the type of concomitant injury [24,25,26]. All patients are routinely seen at our outpatient clinic at 6, 12, 26 and 52 weeks postoperatively.

### 2.4. Image Analysis

Image findings were analyzed in consensus by two musculoskeletal radiologists (J.N. with over 10 years and G.C.F. with 4 years of experience) blinded to all clinical information. The images were assessed for the presence and type of pulley lesions defined by discontinuity of the sGHL, the CHL or both as well as displacement of the LHBT (Figure 1 and Figure 2). Furthermore, images were assessed for concomitant injuries including tears and tendinopathy of the rotator cuff, lesions of the glenoidal labrum, bony defects of the glenoid, injury to the surrounding muscles as well as joint effusion. For a detailed description see Table 2. Image quality, diagnostic confidence and the visibility of the of the anatomical landmarks were graded individually and separately by both raters. Gradings were conducted according to a five-point Likert scale (1 = poor, 2 = below average, 3 = fair, 4 = good, 5 = excellent). The following anatomical landmarks were assessed for their visibility: biceps pulley, LHBT, rotator cuff, glenoidal labrum, axillary recess, articular cartilage, and muscle.

### 2.5. Statistical Analysis

Statistics were performed by G.C.F. (5 years of experience in biostatistics) using IBM SPSS, version 25.0 (IBM Corp., Armonk, NY, USA). Descriptive statistics were performed using McNemar’s tests (for binary categorical variables) and paired t-tests (for numeric variables). Additionally, box plots were created for better visualization of the age distribution. All statistical tests were performed as two sided, and a level of significance (α) of 0.05 was used for all tests.

## 3. Results

### 3.1. MR Assessment of Shoulder Pathologies

In total, 33 shoulders with acute anterior shoulder dislocations were assessed in this study. Seventeen (52%) patients were diagnosed with pulley lesions including sixteen (48%) patients with discontinuity of the sGHL and twelve (36%) patients with discontinuity of the CHL. The mean age of patients with pulley lesions identified on MR imaging was 52 ± 12 years and was significantly higher compared to patients without pulley lesions (mean age 44 ± 14 years, *p =* 0.023, Figure 3). Thirty-two (97%) patients showed a typical defect of the antero-inferior labor-ligamentous complex, of whom fourteen (42%) patients had an additional osseous Bankart lesion and twenty eight (88%) patients a Hill-Sachs defect of the proximal humerus (Figure 4). A total of twenty rotator cuff lesions were detected; with seventeen (42%) lesions involving the SSP tendon and three (9%) the SSC tendon. Of the seventeen patients with pulley lesions (sGHL, CHL or combined lesions), sixteen (94%) patients were diagnosed with an additional lesion of the SSP (N = 13) or SSC tendon (N = 3). A total of four (12%) patients demonstrated medial subluxation of the LHBT. The mean age of patients who showed a lesion of the rotator cuff was 54 ± 19 years and was not significantly higher compared to the patients without a lesion of the rotator cuff (mean age 43 ± 17 years, *p =* 0.29, Figure 5). None of the patients exhibited a notable fatty infiltration or atrophy of the rotator cuff muscles. Joint effusion was detected in all 33 patients due to the acute trauma. Detailed description of the pathologies and their frequencies are shown in Table 2.

### 3.2. Operative Treatment and Correlation with MR Imaging

Seven patients were treated conservatively due to absent clinical signs of apprehension and low sportive demand, and eight patients were lost to follow up. Thus, arthroscopic confirmation of the MR imaging findings could be performed in fifteen patients (46%). All other patients underwent diagnostic shoulder arthroscopy including arthroscopic labral repair using a minimum of three suture anchors ± repair of concomitant pathologies. Upon analyzing the intraoperative reports of the eighteen patients undergoing arthroscopy, six patients were diagnosed with an acute lesion of the biceps pulley and twelve patients did not show injuries to the pulley system. Besides refixation of the ruptured labrum, pulley lesions were treated by either intraarticular or subpectoral tenodesis of the LHBT. When compared to the findings on MR imaging, all six pulley lesions as well as the twelve intact biceps pulleys were accurately identified.

### 3.3. Image Quality, Anatomical Landmarks

The diagnostic confidence was rated high by both raters using a 5-point Likert scale (rater 1: 4.5 ± 0.4, rater 2: 4.3 ± 0.5). Overall, the image quality was excellent and sufficient to assess all acute pathologies (rater 1: 4.3 ± 0.7, rater 2: 4.5 ± 0.4). The visibility of the anatomical landmarks was overall excellent (rater 1: 4.1 ± 0.6, rater 2: 4.4 ± 0.8) (Table 3 and Table 4).

## 4. Discussion

In this study, we assessed the prevalence of injuries to the biceps pulley as well as to the LHBT and rotator cuff by means of MR imaging after acute shoulder dislocation in 33 consecutive patients. Up to 52% of the patients assessed in our study showed a lesion of the biceps pulley following anterior shoulder dislocation and of those, up to 94% showed accompanying pathologies of either the SSC or SSP tendon. In 54% of the cases, arthroscopic reports were available for confirmation and all arthroscopic results were consistent with the MR imaging findings.

The association between pulley lesions and rotator cuff tears has been shown by several studies [8,11,30,31]. Previously, Braun et al. showed that pulley lesions had a prevalence of up to 32% in a large cohort of over 200 patients with anterior shoulder pain and found a significant correlation between pulley lesions and subluxation/dislocation of the LHBT, injuries of the rotator cuff and SLAP lesions [1]. Similar observations were made by Martetschläger et al. who classified lesions of the biceps pulley into type 1 to 3 and was able to detect significant correlations between the type of pulley defect and tear of either the SSC or SSP tendon [2]. In our study, we detected a higher incidence of SSP lesions (N = 13) in patients with pulley lesions compared to SSC lesions (N = 3). This might be due to the fact that, in contrast to previously mentioned studies, we assessed a cohort with traumatic anterior shoulder dislocations and the higher incidence might be due to the different trauma mechanisms. MR arthrography is the current gold standard for the MR imaging of pulley lesions [32]. In this study, we used joint effusion after acute trauma as a natural contrast medium to assess the joint structures and were able to assess the biceps pulley, rotator cuff tendons and labrum with high ratings of diagnostic confidence and visibility of the anatomical landmarks. To the best of our knowledge, the aforementioned results have not been reported yet and raise further awareness of lesions of the biceps pulley in acute traumatic shoulder dislocation, particularly in patients over 40 years of age. This is of particular importance as potential late complications to the long head of the biceps such as tendinitis, irritation, subluxation or even luxation could be averted early. In our study, patients with biceps pulley lesions following anterior shoulder dislocation were significantly older compared to patients without pulley lesions. However, similar to tendon deterioration with increasing age, the components of the biceps pulley are prone to age-related degeneration as well [33]. Hence, whether the presented biceps pulley lesions in older patients may be secondary due to pre-existing intrinsic degeneration or acute damage due to trauma is a matter of debate. Although no statistical significance was reached, we found that patients with rotator cuff lesions were also markedly older than the patients without rotator cuff lesion. An increased prevalence of rotator cuff diseases and increased damage to the rotator interval after dislocation in the elderly has been previously reported [17,34].

There are certain limitations of this study which need to be addressed. First, the patient group was relatively small, with only 33 patients included in the study of whom 17 patients presented with a biceps pulley lesion. However, to the best of our knowledge, it was the largest prospectively recruited cohort with acute shoulder dislocations assessed for biceps pulley lesions. Second, arthroscopic confirmation of pulley lesions diagnosed on MR imaging was only available for 54% of the patients. Seven patients were treated conservatively due to a low sportive demand and age of >40 years, respectively [26]. Finally, pre-traumatic MR imaging was not available to rule out potentially pre-existing shoulder pathologies/tendon degeneration. However, in accordance with the negative clinical history of shoulder pain or previous trauma and absence of atrophy/fatty infiltration of rotator cuff muscles on MR imaging, we believe that our findings are most likely a result of acute anterior shoulder dislocation.

## 5. Conclusions

In this study, a high prevalence of the biceps pulley and rotator cuff lesions was detected in patients following traumatic anterior shoulder dislocation. The aforementioned pathologies might be a frequently unrecognized concomitant injury after traumatic shoulder dislocation. Hence, our results suggest an increased awareness for lesions of the biceps pulley in acute traumatic shoulder dislocation, particularly in patients over 40 years of age.

## Figures and Tables

**Figure 1 diagnostics-12-02345-f001:**
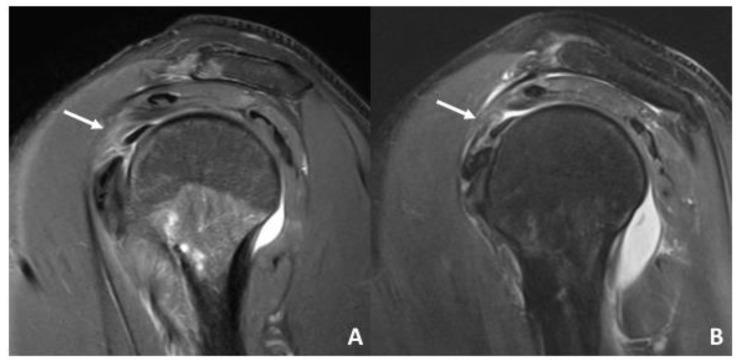
Sagittal IM-weighted TSE sequence with spectral fat saturation. (**A**) Intact biceps pulley in a 26-year-old patient after anterior shoulder dislocation with normal discrimination of the sGHL (white arrow) and CHL. (**B**) Insufficient biceps pulley on a 32-year-old patient after shoulder dislocation due to a tear of the sGHL (white arrow). Caudal displacement of the LHBT is seen at the middle of the rotator interval. Note the joint effusion in both patients which causes an arthrographic effect and helps with the discrimination of anatomical structures.

**Figure 2 diagnostics-12-02345-f002:**
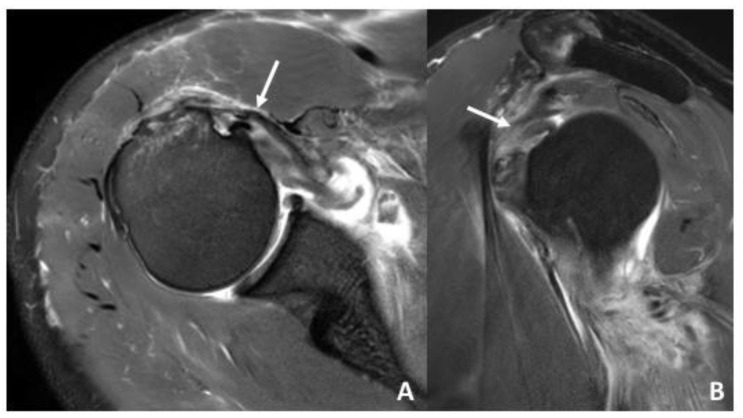
(**A**) Transversal IM-weighted TSE sequence with spectral fat saturation of a 45-year-old patient after traumatic shoulder dislocation with antero-inferior labral lesion and concomitant medial subluxation of the LHBT (white arrow) due an insufficient biceps pulley and a partial SSC tendon tear. (**B**) Sagittal IM-weighted TSE sequence with spectral fat saturation. Pulley lesion in a 39-year-old patient after traumatic anterior shoulder dislocation with tear of the sGHL and CHL. Note the displacement of the LHBT (white arrow) as well as the tendinopathic changes which might be due to preexisting degenerative changes.

**Figure 3 diagnostics-12-02345-f003:**
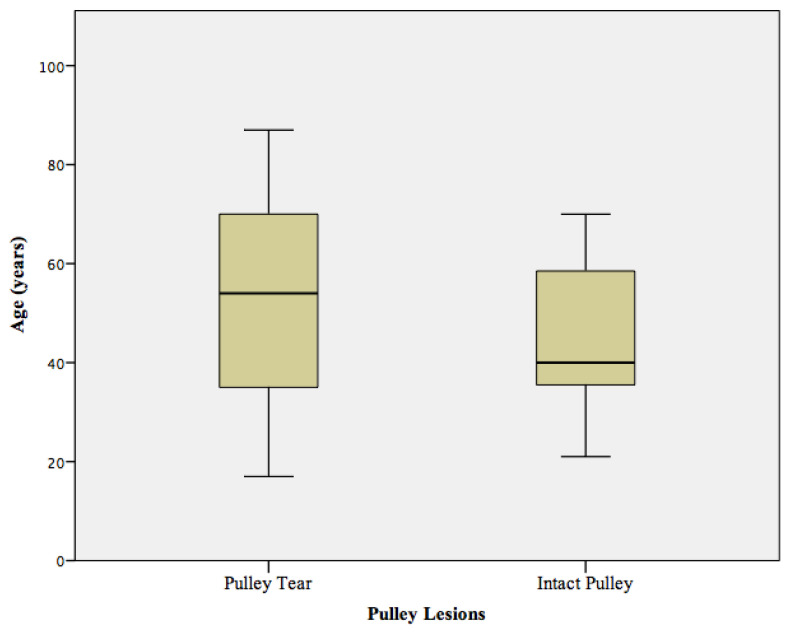
Mean age of patients with acute lesions of the biceps pulley (mean age 52 ± 12 years) compared to patients with intact biceps pulley (mean age 44 ± 14 years, *p =* 0.023). The mean age of patients with acute pulley lesions was significantly higher compared to patients without pulley lesions (*p =* 0.023).

**Figure 4 diagnostics-12-02345-f004:**
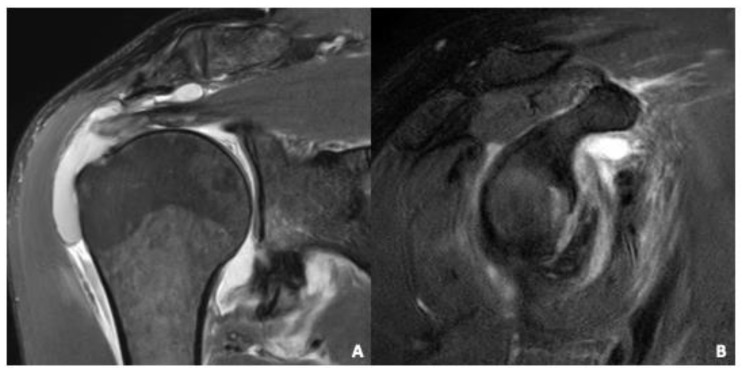
Concomitant injuries after acute anterior shoulder dislocation. (**A**) Coronal IM-weighted TSE sequence with spectral fat saturation. Complete tear of the proximal SSP tendon. Note the markedly increased joint effusion which fills the joint capsule. (**B**) Sagittal IM-weighted TSE sequence with spectral fat saturation. Bony Bankart Lesion with avulsion fracture of the anterior glenoid.

**Figure 5 diagnostics-12-02345-f005:**
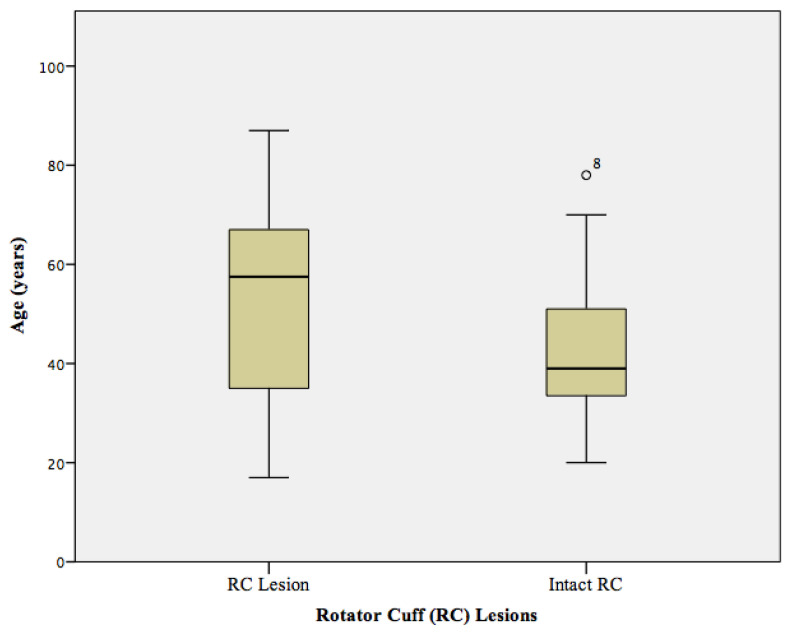
Mean age of patients who showed a lesion of the rotator cuff which was 54 ± 19 years and was not significantly higher compared to the patients without a lesion of the rotator cuff (mean age 43 ± 17 years, *p =* 0.29).

**Table 1 diagnostics-12-02345-t001:** Sequence parameters of the sequences used in this study (Ingenia, Elition, Philips).

Sequence	Axial IM/SPIR	Sagittal IM/SPIR	Coronal IM/SPIR	Sagittal T2	Coronal T1
Echo time (ms)	50	50	50	80	19
Repetition time (ms)	2450	2400	2400	2500	730
TSE factor	16	16	16	16	5
Field of view (mm^3^)	160 × 160 × 83	160 × 160 × 83	160 × 160 × 83	160 × 160 × 108	160 × 160 × 76
Voxel size (acquisition, mm^3^)	0.4 × 0.54 × 3.0	0.4 × 0.54 × 3.0	0.4 × 0.54 × 3.0	0.35 × 0.49 × 3.0	0.35 × 0.45 × 3.0
Voxel size (reconstructed, mm^3^)	0.28 × 0.28 × 3.0	0.28 × 0.28 × 3.0	0.28 × 0.28 × 3.0	0.24 × 0.24 × 3.0	0.24 × 0.24 × 3.0
Slice Thickness (mm)	3	3	3	3	3
Slice number	31	28	26	30	23
Acquisition time (min)	3.24	3.18	3.39	3.11	2.3

**Table 2 diagnostics-12-02345-t002:** Assessed pathologies in patients with traumatic dislocations of the shoulder.

Parameters	Description	Grading and Frequency (n, %)
Biceps pulley	Pulley Lesions graded according to Habermeyer et al. [8]	sGHLtear: 16 (48%) CHL tear: 12 (36%) sGHL and CHL tear: 11 (33%)
LHBT	Subluxation/dislocation of the LHBT	Subluxation: 4 (12%) Dislocation: 0 (0%)
Rotator cuff tendons	Tendinopathy including: subacromial impingement, rotator cuff tendinitis/tendinosis, calcific tendonitis.	No pathology: 10 (30%) Tendinopathy: 16 (48%) Partial or complete tear: 20 (61%)
Rotator cuff muscles	Fatty infiltration assessed according to Goutallier et al. [27]	No pathology: 24 (73%) Fatty infiltration: 0 (0%) Acute injury: 9 (27%)
Bone	Assessment for osseous defects such as osseous/bony Bankart and Hill-Sachs lesions [28]	Normal signal: 1 (63%) Greater tuberosity fracture: 2 (5%) Hill Sachs defect: 29 (88%), Osseous/Bony Bankart defect: 14 (42%)
Labrum	Assessment for defects of the labrum e.g., antero- inferior after dislocation, anatomical variations of the labrum according to Kanatli et al. [29]	Normal Labrum: 1 (3%) Anatomical normal variant: 0 (3%) Lesion: 32 (97%)
Cartilage	Visual grading of signal inhomogeneities or defects of articular cartilage	Normal signal: 25 (76%) Abnormal: 8 (24%)

**Table 3 diagnostics-12-02345-t003:** Mean diagnostic confidence, image quality and visibility.

	Rater 1	Rater 2
Diagnostic confidence ^1^	4.5 ± 0.4	4.3 ± 0.5
Overall image quality ^2^	4.3 ± 0.7	4.5 ± 0.4
Overall visibility of anatomical landmarks ^3^	4.1 ± 0.6	4.4 ± 0.8

Data are presented as means ± standard deviations. The 5-point Likert scale (1 = poor, 2 = below average, 3 = fair, 4 = good, 5 = excellent); ^1^ Diagnostic confidence: describes the level of certainty of the raters regarding the diagnosed pathologies. ^2^ Image quality ratings include: motion artifacts, images noise and general impression. ^3^ Visibility of anatomical landmarks was assessed by rating clearness of the evaluated area and overlying motion or noise artifacts.

**Table 4 diagnostics-12-02345-t004:** Visibility of anatomical landmarks of the shoulder ^1^.

Anatomical Regions	Rater 1	Rater 2
Biceps pulley	4.4 ± 0.4	4.5 ± 0.3
LHBT	4.0 ± 0.4	4.0 ± 0.4
Rotator cuff	4.2 ± 0.3	4.2 ± 0.3
Labroligamentous complex	4.5 ± 0.3	4.5 ± 0.4
Articular cartilage	4.7 ± 0.3	4.7 ± 0.5
Axillary recess	4.0 ± 0.5	4.1 ± 0.5
Inferior glenohumeral ligament	4.1 ± 0.7	4.2 ± 0.2
Bone	4.2 ± 0.4	3.9 ± 0.5
Muscle	4.3 ± 0.3	4.3 ± 0.4

Data are presented as means ± standard deviations. The 5-point Likert scale (5 = best; 1 = worst). ^1^ Visibility was assessed by rating clearness of the evaluated area as well as overlying motion or noise artifacts.

## Data Availability

The data presented in this study are available on request from the corresponding author. The data are not publicly available due to privacy of the study participants.

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
