# Peer review of "Assessment of Acute Lesions of the Biceps Pulley in Patients with Traumatic Shoulder Dislocation Using MR Imaging"

_diagnostics, 2022, doi:10.3390/diagnostics12102345_

Round 1

Reviewer 1 Report

The present study is short on innovation to some extent and the sample size is relatively small. The results do not reflect the important clinical significance and value.

Reviewer 2 Report

The manuscript deals with a very interesting, until now not sufficiently described pathology of traumatic anterior shoulder dislocation: trauma-associated lesions of the biceps pulley. According to this manuscript, lesions of the biceps pulley can be detected in up to 50% and should be recognised more frequently. The manuscript is well structured and well presented. The results are documented sufficiently. The discussion is both, comprehensive and detailed. The tables are well structured. The figures are nicely selected. Only the white arrows are not mentioned/explained in the reference. Overall I congratulate to this original and very informative study.

Round 2

Reviewer 1 Report

After careful consideration, I still feel that the manuscript is not suitable for publication in the current form.

Author Response

Dear Reviewer,

thank you for your comment. We are truly sorry that the previous revision could not fully remove your concerns regarding this study. Nevertheless, we think that this study is worth being published as it raises awareness of the high prevalence of the biceps pulley and rotator cuff lesions in patients following traumatic anterior shoulder dislocation which might avert future complications of underdiagnosed pathologies.

Sincerely,

Georg C. Feuerriegel